# NeuroCORD: A Language Model to Facilitate COVID-19-Associated Neurological Disorder Studies

**DOI:** 10.3390/ijerph19169974

**Published:** 2022-08-12

**Authors:** Leihong Wu, Syed Ali, Heather Ali, Tyrone Brock, Joshua Xu, Weida Tong

**Affiliations:** 1National Center for Toxicological Research, Food and Drug Administration, 3900 NCTR Rd., Jefferson, AR 72079, USA; 2Department of Internal Medicine, University of Arkansas for Medical Sciences, 4301 West Markham, Little Rock, AR 72205, USA; 3Department of Mathematics and Computer Science, University of Arkansas at Pine Bluff, 1200 University Drive, Pine Bluff, AR 71601, USA

**Keywords:** text mining, language model, BERT model, neurological disorders, COVID-19, information retrieval, machine learning

## Abstract

COVID-19 can lead to multiple severe outcomes including neurological and psychological impacts. However, it is challenging to manually scan hundreds of thousands of COVID-19 articles on a regular basis. To update our knowledge, provide sound science to the public, and communicate effectively, it is critical to have an efficient means of following the most current published data. In this study, we developed a language model to search abstracts using the most advanced artificial intelligence (AI) to accurately retrieve articles on COVID-19-associated neurological disorders. We applied this NeuroCORD model to the largest benchmark dataset of COVID-19, CORD-19. We found that the model developed on the training set yielded 94% prediction accuracy on the test set. This result was subsequently verified by two experts in the field. In addition, when applied to 96,000 non-labeled articles that were published after 2020, the NeuroCORD model accurately identified approximately 3% of them to be relevant for the study of COVID-19-associated neurological disorders, while only 0.5% were retrieved using conventional keyword searching. In conclusion, NeuroCORD provides an opportunity to profile neurological disorders resulting from COVID-19 in a rapid and efficient fashion, and its general framework could be used to study other COVID-19-related emerging health issues.

## 1. Introduction

COVID-19 continues to be associated with many severe outcomes, among them neurological disorders (e.g., neurological symptoms and manifestations, neuropsychiatry disorders, neurobehaviors, and neurocognitive observations) which have raised significant public health concerns [1,2,3,4]. Besides the biological association of the COVID-19 virus with neurological disorders, the lengthy timeframe of this pandemic and its associated quarantine add additional layers of stress and loneliness to people, which may lead to neurological disorders. During the pandemic, hundreds of thousands of publications related to COVID-19 have been released in the public domain [5], and it is challenging for neurological scientists to target and read relevant literature to develop scientific solutions for addressing COVID-19-related neurological disorders. The rapid advancement of natural language processing (NLP) technologies with artificial intelligence (AI) may offer a means of mining neurological disorder-related articles rapidly and efficiently to facilitate scientific research, communication with the public, and policy making.

AI-powered NLP has played a significant role in various text mining tasks, including information retrieval, translation, and sentiment analysis. Recently, deep learning and attention-based language models, such as transformers [6] and BERT [7], have revolutionized the field of text mining [8,9,10]. By taking full advantage of big data and high-performance computing capability, the attention models have significantly advanced many text mining tasks. Another significant benefit of these attention models is the enabling of transfer learning: a pre-trained attention model can serve as “basic knowledge” to be further refined to a specific task and can save the model training significant efforts and time by not needing to learn from scratch [11]. In other words, a language model that has been trained on broad scientific literature collected in large databases such as PubMed [10] can be used as a starting point for further tuning for the literature in a specific field.

To take advantage of AI-powered NLP and attention-based models, our group has established a research project named BERTox, which is an NLP framework for causality assessment, question answering, sentimental analysis, and text summarization [12]. The BERTox project aimed to address several specific research needs in predictive toxicology including drug labeling [13], pharmacovigilance [14], and drug-induced liver injury (DILI) [15].

In this study, we applied AI to analyze the relationship between neurological toxicity and COVID-19. Particularly, we developed a language model named NeuroCORD to extract neurological disorder-related articles from the CORD-19 dataset [5]. To mimic a real-world application, this model was trained on literature published before 2020 and then validated with articles published after 2020 in terms of their relevance to COVID-19-related neurological disorders. In the results section, we describe how the model architecture and parameters were optimized. In addition, to find the baseline information retrieval performance, we also compared our proposed model with a keyword search and baseline tests. Finally, the impact on performance was examined and discussed for the modeling factors such as data input type and summarization approach.

## 2. Materials and Methods

### 2.1. CORD-19 Datasets

The COVID Open Research Dataset (CORD-19) consists of a body of comprehensive, machine-readable coronavirus-related literature for data mining [5]. We retrieved 529,651 articles from CORD-19 on 20 April 2021 for this study. Based on the time of publication, approximately 430,000 and 100,000 articles were published before and after 1 January 2021, respectively. To mimic a real-world application, we used articles published before 2021 as the training dataset for building language models and used articles published after 2021 as testing dataset.

### 2.2. Predictive Modeling

A predictive model was developed to parse neurological disorder-relevant papers (i.e., positive) from irrelevant papers (negative). By the consensus of our literature reviewers, the “positive” dataset of papers was determined by relevant keywords and two sets of keywords were compiled to label neurological disorder-related articles from all retrieved CORD-19 articles. In detail, six (6) terms related to neurological disorders (Neurological disorder, Neurological manifestation, Neurological symptom, Neuropsychiatry disorder, Neurobehavior, and Neurocognitive observation) and four (4) terms related to COVID-19 (COVID, SARS, COV-19, and COV19) were applied to the search. To be labeled as relevant/positive, an article must have contained at least one neurological disorder-related keyword and one COVID-19-related keyword. We found 1691 abstracts that were considered positive. To get a balanced training dataset, another 1691 abstracts that did not contain term “Neuro” were randomly selected to form a negative dataset. The final dataset combined from these positive and negative sets and was further divided into training and testing datasets based on publication dates. The resulting 2561 articles published on or before 31 December 2020 were used for training, and the 819 articles published after that date were used for testing. We opted out of classifying the remaining two (2) since these had no publication dates. The training and testing literature datasets can be found in Appendix A.

The input data of NeuroCORD is the abstract of each article. We examined the impact of different input data such as title and full text later during the discussion. In addition, we applied a summarization approach to shorten the abstract into a few key sentences, but no performance improvement was observed when summarization approach was applied (see discussion for more details).

Next, we converted sentences to numerical vectors using a word embedding algorithm. Popular word embedding approaches include one-hot converting, word2vec [16], ELMo [17], and others. Sentence embedding was mostly based on word embedding plus a follow-up approach to integrate the word embeddings into the sentence level. In this study we used the Python sentence-BERT library [18] with three different kernels: allenai-specter [19], roberta-large [20], and glove_6B_300d [21]. The allenai-specter was trained on scientific citations and can be used to find similar papers; the roberta-large model was optimized for semantic text similarity; and the glove_6B_300d model was trained on 6 billion of tokens from WIKI. Overall, allenai-specter, roberta-large, and glove_6b_300d models generated 768, 1024, and 300 features for the input texts, respectively.

After the context features were generated, three modeling algorithms, K-Nearest Neighbors (KNN), Random Forest (RF), and Neural Networks (MLP), were adopted to develop predictive models, using the python sci-kit learn package [22]. An Optuna-based optimization approach was applied to optimize model parameters [23], utilizing the define-by-run principle. Specifically, the user defined an objective function that contained a parameter search space and then ran a study that tested different combinations within that specific parameter space. Pruning algorithms were used to narrow the search for combinations that yielded the most promising results. All computational experiments used Python (version 3.7) in an in-house high-computing cluster environment.

## 3. Results

### 3.1. Study Workflow

The overall study pipeline for developing NeuroCORD is depicted in Figure 1. The first step was to retrieve article data from the CORD-19 dataset and then use the abstracts for the rest of the analysis. Next, we labeled the abstracts with keywords and built a dataset with balanced positive and negative data samples. Next, we split and developed training and testing sets based on article publication dates: articles published after 1 January 2021 and not used for either the training or testing sets were categorized into the external set. After data processing, the final step was to develop and train the NeuroCORD model with summarization, word-embedding, and predictive model algorithms.

### 3.2. Optimizing Modeling Algorithms and Embedding Approaches

We evaluated the impacts of embedding and modeling algorithms using a grid optimization approach. Our evaluation included three embedding algorithms and three modeling algorithms (see Methods) for a total of nine combinations. As shown in Table 1, the allenai-specter + MLP combination delivered the best performance of all the combinations. The results also indicated that allenai-specter and MLP were the best choices for their respective categories in this study.

After determining the best embedding and modeling algorithms, we tuned the MLP model using Optuna. The parameter spaces were defined as follows: the number of hidden layers which varied between one and four, with 10 to 150 units per layer; the activation function was either ReLU (i.e., Rectified Linear Units) or logistic; the learning rate was either constant or adaptive; and the number of iterations was between 100 and 300. Generally, we found that the tuning made little difference to the overall accuracy of the model (See Appendix A); thus, using the default hyperparameters was sufficient.

### 3.3. Manual Model Validation

Of the 819 testing articles, 770 (338 true positives, TP and 432 true negatives, TN) were correctly predicted with 17 false positives (FP) and 32 false negatives (FN). The overall accuracy of the testing dataset was 0.94, and the recall and precision rates were 0.93 and 0.96, respectively.

Since the reviewers were more interested to see whether the model can find more potentially relevant articles, we closely examined these 17 FP articles to understand why they were predicted to be relevant articles by the model, even though they did not contain pre-defined keywords (i.e., labeled as negative). We manually reviewed these 17 FP articles and found at least seven (7) of them were neurological disorder-related COVID-19 studies. For example: Tam et al. [24] reported COVID-19 causes loneliness leading to increased levels of stress in patients with pre-existing neurological diseases (dementia) and their caregivers. Basu et al. [25] reported a COVID study in children on Kawasaki disease (KD), which is a vasculitis that can cause serious neurological issues. Wang et al. [26] reported a meta-analysis to show that 6.7% of pediatrics patients developed nervous system symptoms caused by COVID-19. Doufik et al. [27] reported a case study of two patients who developed acute psychotic disorders with a delusional theme related to the COVID-19 pandemic. Tzur-Bitan et al. [28] showed individuals with schizophrenia, a mental health disease, may have a higher risk for COVID-19 morbidity. Aziz et al. [29] reported a meta-analysis on olfactory dysfunction, a neurological disorder, in patients with COVID-19. Bartrés-Faz et al. [30] reported a study of COVID-19 causing effects on mental health (i.e., loneliness). In all, we demonstrated that our predictive model, based on language modeling, could provide new insights to find articles that could not be found by a simple keyword-based search.

### 3.4. Baselines

Two baseline tests were applied and compared to the NeuroCORD results. The first test (B1) involved randomly shuffling the document labels. Specifically, the positive/negative labels of articles in the training dataset were randomly assigned but kept the same positive/negative ratio. The second (B2) applied generic keywords, such as “study” instead of neurological disorder keywords, to label positive and negative articles; however, the COVID keywords were still applied in B2. The resulting 15,663 articles containing “study” and COVID-19 keywords in their titles were selected and a subset of 1691 articles from these 15,663 articles was used as positive samples in the training dataset. To generate a balanced dataset, another 1691 articles were picked as negative samples in the training dataset. Thus, we generated a corresponding testing set with articles published after 2021.

The results of the baseline tests and their comparison to the original analyses are summarized in Table 2 for the combination of allenai-specter and each modeling algorithm. The average performance of B1 was approximately 0.5 for both training and testing results. This result implied that all three modeling algorithms were valid and there was no information leaking in our training and testing workflow. The B2 test yielded approximately 0.775 and 0.722 accuracy for training and testing, respectively. B2’s weaker performance demonstrated that the separation of COVID-19-related neurological disorder-related COVID-19 articles and others was clear since the model better predicted neurological disorder articles than a weaker labeled dataset.

### 3.5. Clustering Analysis of Neurological Disorder-Related Articles

NeuroCORD achieved high accuracy (i.e., 0.94) on external testing dataset and reliability in finding COVID-19-related neurological disorder articles. Could its high accuracy lead to the identification of more COVID-19 research articles relevant to the study of neurological disorders? NeuroCORD was then applied to all publications in 2021 to determine whether it could find more neurological disorder-related articles, even if they had been labeled as negative based on the keyword-based query. Overall, 97,311 articles were both published and collected in 2021 by the CORD-19 dataset; among them, 819 had already been used as testing datasets, but the remaining 96,492 were not used in the previous model training or testing process. In total, 3048 articles were predicted by NeuroCORD to be positive, meaning that they are likely relevant to neurological disorder research. Assuming all of these were predicted correctly, the real detection rate would become ~3.5%.

T-SNE analysis was then applied to the training dataset (2561), and all 2021 publications (97,311), which were made of the testing set and the external set. As shown in Figure 2, (a) is the t-SNE distribution of the training dataset, (b) is the distribution of new publications, and (c) is the combination of both. (Predicted) positive articles were highlighted in blue, red, and purple in three sub-figures and negative articles were gray. We observed that most positive articles in the t-SNE result formed a cluster, which could potentially represent a hot spot of neurological disorder articles.

Through a closer examination of the cluster with an applied bounding box [−45, −30, −10, 0] as shown by the red rectangle in Figure 2a, we found 960 of 994 (96.7%) of the articles inside this bounding box were positive. Further investigations of this cluster revealed that most articles contained such keywords as “COVID”, “SARS”, “neurological”, “infection”, “symptoms”, “severe”, etc., indicating that this cluster was highly enriched with neurological disorder and COVID-19 keywords. By applying the same bounding boxes to the t-SNE plot for the new publications, we also found the same keyword distribution and enrichment. Conversely, an application of another bounding box [20, 30, 0, 10] as shown by the green rectangle in Figure 2a, yielded keywords enriched in this cluster of “health”, “gatherings”, “social”, and “pandemic”, which implied that articles in this cluster were also highly related to the COVID-19 pandemic but more related to population analysis. Simply stated, these papers might be irrelevant to neurological disorders and could be skipped in literature reviews to save time.

### 3.6. Discovering New Keywords and Topics

In addition to a strong prediction performance, NeuroCORD could also help find unspecified neurological disorder keywords that have not been used in dataset labeling, especially from those labeled as irrelevant by the initial set of keywords. For example, some research articles reported studies of COVID-19 patients with neurological disorder pre-conditions, such as schizophrenia [28] and dementia [24], while other relevant articles discussed mental health and stress indirectly caused by COVID-19 virus through the pandemic’s intense psychosocial stress [30]. These articles and their keywords, although different from our pre-defined ones, could provide additional context to researchers to further improve their keyword list to find relevant articles.

## 4. Discussion

We developed NeuroCORD, a language model based on literature abstracts, to classify and separate neurological disorder-related COVID-19 articles from others. The combination of the allenai-specter embedding algorithm and neural networks modeling algorithm performed best in both training and testing results. The proposed model achieved 0.972 and 0.940 accuracy in training and testing datasets, respectively. Furthermore, of the 97,311 new articles published in 2021, 3480 (432 + 3048) articles were predicted to be relevant, reflecting a relevant ratio of 3.58%.

In addition to the overall strong modeling performance, we examined the “false positives” from the model outputs. These articles did not contain any predefined keywords, but the model had predicted that they were relevant. We found many of these “false positives” to be actually relevant and of interest to us. The approach of developing predictive modeling to classify neurological disorder-related articles could also help researchers discover new keywords that may be missing from their predefined lists as well as find other related research directions for their topics.

There are several related works that also used transformer models for literature screening. For example, Qin et al. developed an ensembled language model to screen literature related to sodium-glucose co-transporter-2 inhibitors for type 2 diabetes mellitus treatment [31]. Carvallo et al. conducted a comparative study and demonstrated BERT has state-of-the-art performance over other text mining approaches for document screening [32]. Compared to these previous studies, there are several unique contributions of our study. Our NeuroCORD model used a keyword search result as the initial endpoint which can be generated automatically, therefore dramatically reducing the effort in data preparation and maintaining competitive performance. This study focused on neurological disorders, and we took advantage of the domain experts to evaluate the model performance, in addition to the statistical performance. The feedback of the reviewers was valuable to further improve the model. Our study, to the best of our knowledge, is the first application of a transformer model to meet FDA regulatory scientists’ needs in literature screening. Therefore, this work will help regulatory reviewers in their daily task to screen literature and could be a starting point for follow-up studies.

In the rest of this section, we further examined other modeling factors such as data input type and whether to use a summarization approach. We also addressed the limitations of keyword-based searches in comparison to our NeuroCORD model.

### 4.1. More Investigations on the Data Inputs

Besides using abstracts, we also tried to use titles and article full texts as data inputs. It is worth noting that the CORD-19 json files consisted of two sources of json data; one converted from PubMed Central (PMC) and one using the PDF file. In situations where the json data from both sources were available, we only used the PMC json data due to its fine format. From the resulting 529,651 articles, 529,397 titles were extracted from the meta table and 186,530 full texts were extracted from json files, as summarized in Appendix A. We also performed keyword searches on title, abstract, and full-text data. For instance, at the title level there were 447 relevant articles that included at least one neurological disorder keyword and one COVID-19 keyword in the title. Of the 529,397 articles queried, the relevant ratio was approximately 0.1% (447/529397). Similarly, we found 1691 and 3427 relevant articles at the abstract and full text levels, yielding relevant ratios of 0.5% and 2%, respectively.

As shown in Appendix A, the relevant ratio on the abstract level was five (5) times higher than that of the title level, whereas the ratio on the full text level was even higher. These results indicated that the title may contain insufficient information to determine whether an article was relevant to neurological disorders or COVID-19; thus, using the title only to find articles might exclude some relevant articles. On the other hand, we noticed several downsides to using full texts: first, not all articles contained an available full text; second, pre-processing of full texts was more error-prone than for abstracts due to variations in data formatting. These discrepancies largely affect data quality, especially context information, which will require increased data cleaning. Due to these observations, we chose to use the abstract as the input text level in our study.

### 4.2. Fine-Tuning Input via the Text Summarization Approach

In this study, we directly used the abstract as the input of the text embedding process but also considered using the text summarization approach to process abstracts before text embedding. Text summarization is a typical task of language modeling used to summarize a long document into a short paragraph or sentence [33,34]. There are two different strategies to perform text summarization, as abstractive and extractive, which depend on whether the exact raw texts are used in the output [35]. In this study, we used a Google Pegasus pretrained model [36] to perform text summarization.

To evaluate the performance of text summarization methods, we compared three different approaches: the first was the direct use of the entire abstract as input and generate embedding vectors (as previously presented in this study); the second was a text summarization approach to first convert the abstract into one or two sentences before generating embedding vectors; and the third was to separate the entire abstract at sentence level and then convert each sentence to embedding vectors. We measured the training and testing performances of all three approaches. Except for the input processing, all other parts of the model training process were kept the same. These three approaches were identified as Abstract, Summary, and Sentence in Appendix A.

The allenai-specter + neural networks pipeline was used in all above approaches and the same training/testing split were kept. As the results show in Appendix A, the abstract approach showed the highest training and testing accuracy (0.972 and 0.940) among all three methods, while the performance of sentence level was the lowest. We then combined the prediction results for sentences from the same abstract using a winner vote algorithm, which resulted in an accuracy of 0.921 on the abstract level.

Note that our model, directly using the raw abstract as the input, achieved the best performance; simply stated, neither sentence nor summarization level could further improve the model performance in this study. A potential explanation is that abstracts were already condensed versions of the full texts, further condensing these abstracts could lead to information loss and a reduced performance for the model. Conversely, separation of complete abstracts into sentence levels and assignment of the same label to each sentence might have increased inaccuracy as generic sentences (such as introductory sentences) in relevant articles might have been given relevant labels as well. Finally, we decided to develop the NeuroCORD model based on abstracts directly rather than using text summarization approach.

### 4.3. Limitations of Keyword Search

Although the keyword-based search can find articles with designated relevant keywords, it has several limitations. Considering our research with the keyword search, first, it is essential to have a large list of relevant keywords to extract a comprehensive list of articles on COVID-19-related neurological disorders, which is difficult and can introduce potential false positives. The keyword search method largely relies on matches with standard terminology, where in some cases a small typo could affect the search results, or on controlled vocabulary that incorporates a predetermined list of synonyms or related concepts. Second, it cannot deal with homonyms (words that have multiple meanings), where context is often defined by surrounding words and sentence structure. Alternatively, the attention-based language model can handle both typos and homonyms by taking advantage of semantic context, as demonstrated.

## 5. Conclusions

In summary, we demonstrated that NeuroCORD provides an opportunity to address neurological disorders associated with COVID-19 in a rapid and efficient fashion and the general framework of NeuroCORD can be used to study other COVID-19-related emerging health issues. Our approach can be easily applied to other topics, such as cardiovascular diseases and drug side effects, by changing the initial neurological disorder relevant keywords list.

## Figures and Tables

**Figure 1 ijerph-19-09974-f001:**
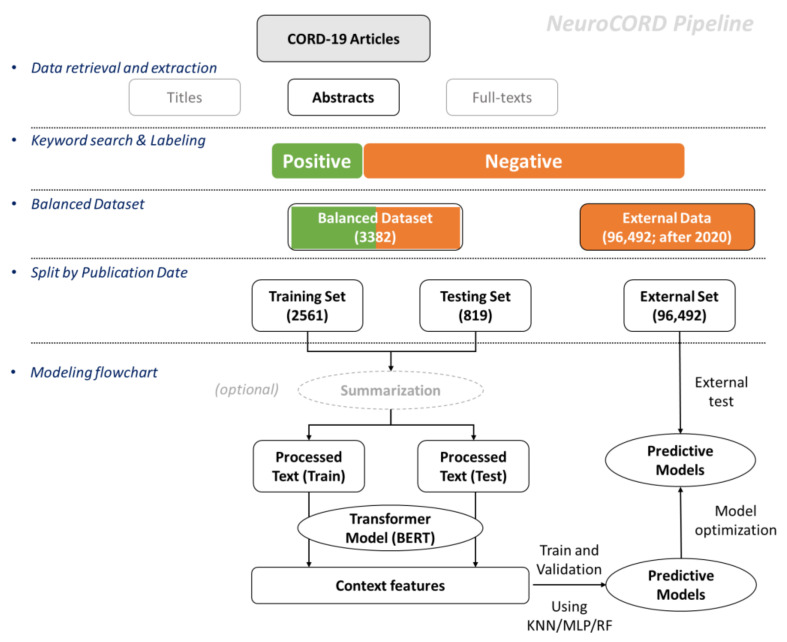
The overall study pipeline for developing the NeuroCORD model.

**Figure 2 ijerph-19-09974-f002:**
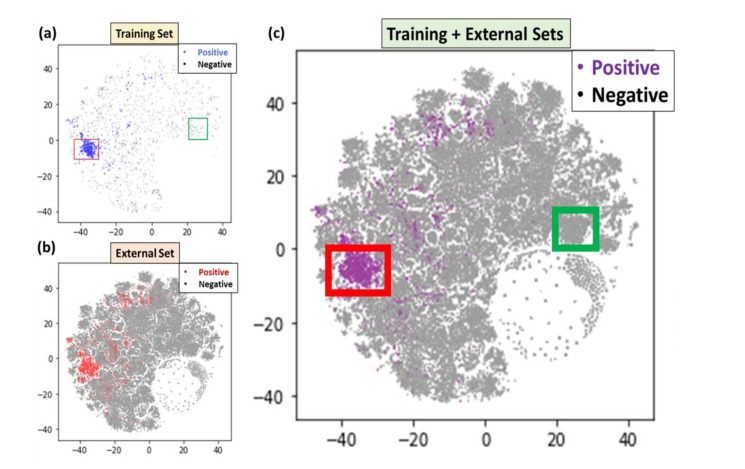
T-SNE analysis of new publications (in 2021). Distribution of articles (**a**) in the training dataset, (**b**) in the new publications, and (**c**) in the combined dataset. (Predicted) positive articles are in blue/red/purple; negative articles are in gray. The red and green bounding boxes in (**a**,**c**) are the two clusters discussed in this paper.

**Table 1 ijerph-19-09974-t001:** Result of grid optimization on embedding and modeling algorithms.

Embedding Algorithm	Modeling Algorithm	10-Fold CV (2561)	Testing (819)
allenai-specter	RF	0.959	0.923
	KNN	0.906	0.904
	MLP	0.972	0.940
roberta-large	RF	0.936	0.855
	KNN	0.839	0.832
	MLP	0.952	0.926
glove_6B_300d	RF	0.930	0.889
	KNN	0.857	0.882
	MLP	0.943	0.940

**Table 2 ijerph-19-09974-t002:** Results of baseline tests.

	NeuroCORD	Test	Train (B1)	Test (B1)	Train (B2)	Test (B2)
**KNN**	0.906	0.904	0.508	0.514	0.698	0.703
**RF**	0.959	0.923	0.500	0.479	0.746	0.709
**MLP**	0.972	0.940	0.499	0.471	0.775	0.722

## Data Availability

Not applicable.

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
