# Peer review of "NeuroCORD: A Language Model to Facilitate COVID-19-Associated Neurological Disorder Studies"

_ijerph, 2022, doi:10.3390/ijerph19169974_

Round 1

Reviewer 1 Report

Since the beginning of COVID-19 many papers have been published, being difficult and overwhelming to find papers related to a specific topic which are relevant. In the light of this circumstances, in this work, the authors developed NeuroCORD, a language model aimed to retrieve articles related with COVID-19-associated neurological disorders.

After a detailed review, I consider that this might be an interesting paper. However, I think that it is out of the scope of the special issue that it was sent to, an even to the area of the IJERPH journal that this special issue belongs.

Please, check my comments in the attached pdf file.

Author Response

Dear reviewer,

thanks very much for your valuable comments, we have revised our manuscript accordingly, please see the attached response letter and the revised manuscript.

best regards

Leihong, on behalf of all authors

Reviewer 2 Report

The paper presents a language model for COVID-19-associated neurological disorder studies using machine learning techniques. The paper needs some modifications before it is published in the journal.

·        The discussion on modeling algorithms is too short (lines 105-11). The authors need to explain how these models are used.

·        In line 127 the authors refer to the same modeling algorithms for details however the details are not available.

·        Summarization and word embedding in figure 1 need discussion.

·        The authors need to justify the choices for training and testing values.

·        In the caption of figure 2, a, b, and c are referred however there are no labels on the figures.

·        Will the data generated in this work be available to other researchers?

Author Response

(The authors gave the same response as above.)

Round 2

Reviewer 1 Report

The authors have replied to all my questions.

Now I consider that it can be accepted for publication.

Reviewer 2 Report

My concerns are addressed. Authors need to include the explanation of the models used. This will improve the quality of the paper.